# A Stronger Mixture of Low-Rank Experts for Fine-Tuning Foundation Models

**Mengyang Sun** [† 1]  **Yihao Wang** [2]  **Tao Feng** [1]  **Dan Zhang** [† 1]  **Yifan Zhu** [3]  **Jie Tang** [1]

## Abstract

In order to streamline the fine-tuning of foundation models, Low-Rank Adapters (LoRAs) have been substantially adopted across various fields, including instruction tuning and domain adaptation. The underlying concept of LoRA involves decomposing a full-rank matrix into the product of two lower-rank matrices, which reduces storage consumption and accelerates the training process. Furthermore, to address the limited expressive capacity of LoRA, the Mixture-of-Expert (MoE) has been introduced for incorporating multiple LoRA adapters. The integration of LoRA experts leads to a visible improvement across several downstream scenes. However, the mixture of LoRAs (MoE-LoRA) still exhibits its low robustness during tuning and inferring. Inspired by the Riemannian Preconditioners which train LoRA as a sub-space projector, we propose a new training strategy for MoE-LoRA, to stabilize and boost its feature learning by gate-rescaled multi-space projections. We provide both a theoretical solution as well as an alternative engineering strategy. Examinations on SGD and AdamW optimizers demonstrate the effectiveness of our methodology. Source code is available at https://github.com/THUDM/MoELoRA_Riemannian.

## 1. Introduction

Parameter-Efficient Fine-Tuning (PEFT) techniques offer a cost-effective solution for fine-tuning foundation models (FMs) (Zhang et al., 2025a; Fu et al., 2024). Among these, Low-Rank Adaptation (LoRA) is a prevalent technology due

to its versatility and simplicity. In detail, LoRA introduces trainable low-rank matrices $A$ and $B$ to update the internal modules of FMs, which is given by $X = W + BA$, where $X$ represents the overall weight matrix after integrating pretrained weights $W$ and LoRA modules $A$ and $B$. In a sense, the product of $A$ and $B$ serves as an approximation of the full-rank update (in other words, Fully Fine-Tuning, or FFT) for the pre-trained weights. While LoRA significantly reduces the number of trainable parameters, it also imposes two limitations: One is limited representation, and the other one is gradient sub-optimality.

***Limitation 1:*** *Limited representation.* A natural problem of low-rank matrices lies in less powerful representation, especially in complex tasks. To tackle this, one straightforward solution is the integration of multiple LoRA modules into the mixture-of-expert framework, known as MoE-LoRA. Figure 1 (left) illustrates a plain MoE-LoRA framework. These efforts tangibly improved the performance of LoRA in many scenarios, like vision-language tasks, multi-task learning, continual learning, etc. In a nutshell, the route of MoE-LoRA can be roughly categorized into two lines: (i) Designing dedicated MoE-LoRA frameworks for specific domains, such as MOELoRA (Liu et al., 2023) and MoCLE (Gou et al., 2023). (ii) Technically improving MoE-LoRA via architectural, updating, and loss constraints, such as MoLA (Gao et al., 2024) and HydraLoRA (Tian et al., 2024). Nevertheless, most of these efforts fail to consider the instability and inefficiency of training MoE-LoRA.

***Limitation 2:*** *Gradient Sub-optimality.* Another concern that plagues LoRA is gradient sub-optimality. This occurs since the low-rank matrices $A$ and $B$ together form a quotient manifold space with a certain curvature, leading to an inconsistency between the inner-manifold optimal and the full-rank optimal gradient. This further leads to a sub-optimal training process for LoRA. To alleviate, Zhang et al. (Zhang & Pilanci, 2024) enhances LoRA gradients by a Riemannian gradient preconditioner, given by $\nabla_A \mathcal{L} = (B^T B)^{-1} \nabla_A \mathcal{L}$ and $\nabla_B \mathcal{L} = \nabla_B \mathcal{L}(AA^T)^{-1}$. These preconditioners contribute to constructing two gradient projectors after a mathematical derivation, ensuring the update is done in accord with the full-rank gradient projection onto the row space of $A$ and the column space of $B$, that is $X_{new} = X - \eta[Proj_{col(B)}(\nabla_X \mathcal{L})^T + Proj_{row(A)}(\nabla_X \mathcal{L})]$. Here the notation $Proj_V(M)$ represents a projection func-

† Work was done while these authors interned at Zhipu AI.

[1]Department of Computer Science and Technology, Tsinghua University, Beijing, China; [2]Computer School, Beijing Information Science and Technology University, Beijing, China; [3]School of Computer Science, Beijing University of Posts and Telecommunications, Beijing, China. Correspondence to: Jie Tang <jietang@tsinghua.edu.cn>, Tao Feng <fengtao.hi@gmail.com>.

*Proceedings of the 42$^{nd}$ International Conference on Machine Learning*, Vancouver, Canada. PMLR 267, 2025. Copyright 2025 by the author(s).

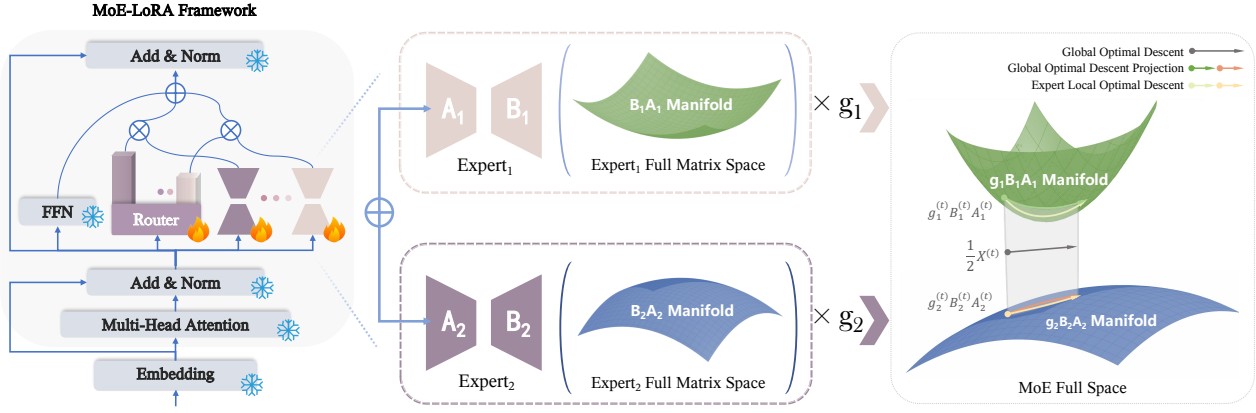

*Figure 1.* The whole MoE-LoRA architecture and an insight into its gradient updating process. The left part of this figure shows a pipeline of mixture of LoRAs, which fixes the pretrained weights of Feed-Forward Network (FFN), and trains a series of LoRA adapters together with a routering gate. The right part exhibits how MoE-LoRA is updated. Specifically, we plot an example of a 2-Expert MoE-LoRA in a condition that $g_1 < g_2$, which results in a further distorted manifold $g_1 B_1 A_1$. Here we simply omit the fixed pretrained weights and suppose $X = g_1 E_1 + g_2 E_2$ for convenient display. Since that, for a random step $t$ we plot a state point $\frac{1}{2} X^{(t)}$, which equals to $\frac{g_1^{(t)} B_1^{(t)} A_1^{(t)} + g_2^{(t)} B_2^{(t)} A_2^{(t)}}{2}$ and so that serves as the center point of the two manifold states at $t$. This figure illustrates that $g_1 B_1 A_1$ has a higher curvature so that its local optimal descent and its global optimal descent projection are more distinct. That indicates a requirement for gate-related preconditioners.

tion which projects a given matrix $M$ onto a subspace constructed by all vectors in set $V$.

Through a comprehensive analysis of *Limitation 1* and *Limitation 2*, a natural question arises:

> *How can LoRA-based structure further approximate the full fine-tuning with the guaranteed Limitations 1 and 2?*

Inspired by MoE-LoRA and the gradient preconditioning methods, a straightforward answer to this question is to integrate both approaches to simultaneously overcome the representative and sub-optimal limitations. Specifically, the gradients of each LoRA expert can be refined by a respective Riemannian preconditioner. However, we claim that the process of weighed summing experts in MoE-LoRA introduces a gate-based scaling for each LoRA expert's manifold, thereby altering their curvatures with regard to their respective gate value $g_i$s. We illustrate this phenomenon in the right part of Figure 1, which plots an example of a 2-Expert MoE-LoRA in a condition that $g_1 < g_2$. Specially, in their respective spaces of Expert 1 and 2, Manifolds constructed by $B_1 A_1$ and $B_2 A_2$ initially share the same curvature since their low-rank matrices are in the same rank. However, after being multiplied by gate values, Manifold $g_1 B_1 A_1$ is more rescaled so that it provides a larger curvature than $g_2 B_2 A_2$ in the MoE full space. As a result, Expert 1 exhibits a higher distinction between global optimal and inner-manifold optimal descents. This phenomenon indicates that the preconditioners for each expert shall be further refined, to take the impact of gate values into consideration. In this paper, we propose a simple but effective solution to further rescale the gradients of each expert in a lightweight way by respective

gate value $g_i$. Our improved gradient updating process for MoE-LoRA is given by:

$$
\begin{aligned}
X_{new} = X - \eta \sum_{i=1}^{N_{Expert}} g_i Proj_{col(B_i)}(\nabla_X \mathcal{L})^T \\
- \eta \sum_{i=1}^{N_{Expert}} g_i Proj_{row(A_i)}(\nabla_X \mathcal{L}).
\end{aligned}
$$

We summarize our contributions as follows:

- We integrate the mixture of LoRAs structure with the Riemannian preconditioners to alleviate both limited representation and sub-optimality issues of LoRA.
- We emphasize a distortion issue behind per-expert preconditioning, and respectively propose a theoretical and an engineering solution for gate-value-rescaled gradient preconditioning of MoE-LoRA.
- We implement and examine our rescaling approach for MoE-LoRA under a series of foundation models, illustrating our effectiveness across various tasks.

## 2. Related Works

### 2.1. LoRA and LoRA Variants

LoRA (Hu et al., 2022) decomposes a full-rank matrix into a product of two low-rank matrices, which has been widely considered an effective solution for parameter-efficient fine-tuning. Studies have proposed several variants to reform LoRA: For initialization, PISSA (Meng et al., 2024) leverages singular value decomposition (SVD) to obtain the principal singular components of $W$, while MiLoRA (Wang

et al., 2025a) utilizes secondary singular values and vectors. LoRA-Pro (Wang et al., 2025b) and LoRA-GA (Wang et al., 2024b) approximate the direction of initial gradients to align them with that of the fully fine-tuning. LoRA+ (Hayou et al., 2024) introduces a learning rate separating strategy with $\eta_B > \eta_A$. ResLoRA (Shi et al., 2024b) and SIBO (Wen et al., 2024) accelerate convergence and mitigate over-smoothing by introducing residual paths. DoRA (Liu et al., 2024b) decomposes the weight vector into direction and magnitude and only uses its direction component. rsLoRA (Kalajdzievski, 2023) proposes a rank-stabilized scaling factor $\lambda_t = r_t^{1/2}$ to ensure stable gradient updates. To prevent overfitting, BiLoRA (Qiang et al., 2024) adopts a bi-level optimizing strategy, while others implement dropout mechanisms (Wang et al., 2024a; Lin et al., 2024).

## 2.2. Mixture of LoRAs

MoE has emerged as a critical framework for addressing complex tasks. By incorporating multiple expert modules, it dynamically selects appropriate experts based on specific inputs (Jacobs et al., 1991). Early studies, such as Lo-RAMoE (Dou et al., 2024) and MixLoRA (Li et al., 2024), have pioneered the introduction of the MoE-LoRA architecture by integrating LoRA experts for both global and downstream tasks. Afterward, MoE-LoRA has demonstrated its effectiveness across a range of fields such as continual learning (Dou et al., 2024; Yang et al., 2024), vision-language multi-model tasks (Gou et al., 2023; Chen et al., 2024), and multi-task applications (Liu et al., 2023).

Recent studies have focused on enhancing MoE-LoRA through architectural advancements and improved training strategies. For instance, MoLA (Gao et al., 2024) allocates a varying number of experts at different layers, and MixDA (Diao et al., 2023) introduces multiple domain-adaptive modules to support multi-domain knowledge. Other methods such as (Wu et al., 2024a; Liu et al., 2023; Wu et al., 2024b; Gou et al., 2023; Wang & Agarwal, 2022) have also been proposed for strengthening MoE-LoRA. To boost the training of MoE-LoRA, Luo et al. (Luo et al., 2024) address the random routing issue by introducing a contrastive loss (Shi et al., 2024a). At the same time, MoV (Zadouri et al., 2024) chooses to combine lightweight vectors with a sparse selection mechanism for efficient expert allocation. Other approaches, including (Dou et al., 2024; Li et al., 2024; Zhu et al., 2023), focus on load balancing among experts. However, to the best of our knowledge, there is still a lack of work on gradient optimizing (Bian et al., 2024) specifically for MoE-LoRA models.

## 2.3. Gradient Preconditioners

In most deep learning cases (Zhang et al., 2024a; Luo et al., 2025a), gradient descent algorithms update model parame-

ters by calculating gradient-based updates. To accelerate the optimizing process, the concept of gradient preconditioning has been introduced. Advanced techniques such as Adagrad (Duchi et al., 2011) dynamically adjust the learning rate by an accumulated squared gradients $G_t = \sum_{i=1}^{t} g_i^2$ and update model by $\Delta\theta_t = -\eta G_t^{-1/2} \cdot g_t$. Adam (Kingma & Ba, 2015) extends this approach by incorporating momentum and bias correction, scaling gradients through a diagonal preconditioner, and resulting in updates in the form of $\Delta\theta_t = -\eta \frac{m_t}{\sqrt{v_t}+\epsilon}$, where $v_t = \beta_2 v_{t-1} + (1-\beta_2)g_t^2$. AdamW (Loshchilov & Hutter, 2019) further introduces a weight decay to Adam.

Recent studies have provided theoretical support for scaled gradient descent methods under different preconditioning strategies. The core idea is to adjust both the direction and magnitude of updates by introducing a scaling matrix to gradients. Tong et al. (Tong et al., 2021) demonstrate the local convergence of scaled gradient descent methods. Jia et al. (Jia et al., 2024) extend this work by proving global convergence of scaled gradient descent for the least-squares matrix decomposition problem $\|AB^T - Y\|_F^2/2$, showing that this approach achieves global convergence under different condition numbers. Other variants of scaled gradient descent have also emerged, such as Zhang et al. who proposed two regularization strategies (Zhang et al., 2023; 2024c). In higher-dimensional settings, scaled gradient descent has been further extended to tensor optimization (Tong et al., 2022; Ma et al., 2024). Mishra et al. (Mishra et al., 2013; Mishra & Sepulchre, 2016) also applied the principles of Riemannian to the optimization involving low-rank matrices. Considering the data's manifold geometry, a Riemannian metric $g_p(v, w)$ is introduced to guide gradient updates along the manifold. Recently, Zhang et al. (Zhang & Pilanci, 2024) introduced the idea of Riemannian preconditioners to LoRA by attaching an $r \times r$ preconditioner to the gradients of low-rank matrices. As a result, they provide improved fine-tuning performance of LoRA, compared with conventional gradient optimizers such as SGD and AdamW.

## 3. Method

We elaborate on our motivations and detail the modification we have made to the Riemannian preconditioning method specifically for MoE-LoRA. Our theoretical foundations and engineering solutions are also presented.

### 3.1. Riemannian Preconditioner in LoRA Expert

As a preliminary, we first briefly introduce the Riemannian preconditioner (Zhang & Pilanci, 2024). Suppose the pretrained model weight is $W$ and its additive low-rank components as $B$ and $A$, let $X = W + BA$ denote the whole weight matrix and let $\mathcal{L}$ and $\eta$ denote the loss function and

the learning rate, respectively. For the plain gradient descent method, the gradient updating process is described through Equation (1) to (4), in which the derivation from (2) to (3) relies on ignoring the second-order term of learning rate. Obviously, $B\nabla_A\mathcal{L} + \nabla_B\mathcal{L}A$ in (4) serves as an approximation of the ideal FFT gradient of $X$.

$$X_{new} = W + B_{new}A_{new} \quad (1)$$
$$= W + (B - \eta\nabla_B\mathcal{L})(A - \eta\nabla_A\mathcal{L}) \quad (2)$$
$$\approx W + BA - \eta B\nabla_A\mathcal{L} - \eta\nabla_B\mathcal{L}A \quad (3)$$
$$= X - \eta(B\nabla_A\mathcal{L} + \nabla_B\mathcal{L}A) \quad (4)$$

Subsequently, according to the derivation chain rule and the simple fact that $X = W + BA$, we directly obtain that $\nabla_A\mathcal{L} = (\nabla_A X)(\nabla_X\mathcal{L}) = B^T(\nabla_X\mathcal{L})$, and likewise $\nabla_B\mathcal{L} = (\nabla_X\mathcal{L})A^T$. Thus, (4) can be transformed to:

$$X_{new} = X - \eta[BB^T(\nabla_X\mathcal{L}) + (\nabla_X\mathcal{L})A^TA], \quad (5)$$

which actually updates the model in a different direction compared to the FFT update formula $X_{new} = X - \eta\nabla_X\mathcal{L}$. This phenomenon occurs since the distorted sub-space of $X$ constructed by $BA$ brings inconsistency between the optimal gradient descent within its manifold and that of the full matrix $X$. To address this inconsistency, Zhang et al. (Zhang & Pilanci, 2024) scale the gradients of $A$ and $B$ by:

$$\nabla_A\mathcal{L} = (B^TB)^{-1}\nabla_A\mathcal{L}$$
$$\nabla_B\mathcal{L} = \nabla_B\mathcal{L}(AA^T)^{-1}, \quad (6)$$

so that (5) is expressed as:

$$X_{new} = X - \eta[B(B^TB)^{-1}B^T(\nabla_X\mathcal{L})$$
$$+ (\nabla_X\mathcal{L})A^T(AA^T)^{-1}A]$$
$$= X - \eta[Proj_{col(B)}(\nabla_X\mathcal{L})^T$$
$$+ Proj_{row(A)}(\nabla_X\mathcal{L})], \quad (7)$$

where the update inside the manifold is performed according to the full matrix gradient projection onto the row space of $A$ and the column space of $B$. Therefore, it better approximates fully fine-tuning than the unscaled descent step.

Inspired by this work, a straightforward way to expand their solution to MoE-LoRA is to individually scale the gradient of each LoRA expert by (6). However, equation $X = W + BA$ lays out in a different form in MoE-LoRA:

$$X = W + \sum_{i=1}^{N_{Expert}} g_iB_iA_i, \quad (8)$$

where $N_{Expert}$ denotes the number of activated experts and $g_i$ denotes the gate value of specific expert $i$. As a result,

it not only brings a gate value $g_i$ for each expert $i$ into Equation (1)-(4), but also introduces an extra gate value $g_i$ for each expert $i$ into (5), since the derivation chain rule $\nabla_{B_i}\mathcal{L} = g_i(\nabla_X\mathcal{L})A_i^T$ and $\nabla_{A_i}\mathcal{L} = g_iB_i^T(\nabla_X\mathcal{L})$. To further clarify, we formally derive the whole result. Note that gate values are computed through a softmax with complex non-linear operations, thus we just treat them as constants for an easier deriving approximation. Following the conventional Riemannian preconditioners in (6), we have:

$$X_{new} = W + \sum_{i=1}^{N_{Expert}} g_i(B_i - \eta\nabla_{B_i}\mathcal{L})(A_i - \eta\nabla_{A_i}\mathcal{L})$$
$$\approx X - \eta\sum_{i=1}^{N_{Expert}} g_i(B_i\nabla_{A_i}\mathcal{L} + \nabla_{B_i}\mathcal{L}A_i)$$
$$= X - \eta\sum_{i=1}^{N_{Expert}} g_i[B_i(B_i^TB_i)^{-1}\nabla_{A_i}\mathcal{L}$$
$$+ \nabla_{B_i}\mathcal{L}(A_iA_i^T)^{-1}A_i] \quad (9)$$
$$= X - \eta\sum_{i=1}^{N_{Expert}} g_i[g_iB_i(B_i^TB_i)^{-1}B_i^T(\nabla_X\mathcal{L})$$
$$+ g_i(\nabla_X\mathcal{L})A_i^T(A_iA_i^T)^{-1}A_i]$$
$$= X - \eta\sum_{i=1}^{N_{Expert}} g_i^2Proj_{col(B_i)}(\nabla_X\mathcal{L})^T$$
$$- \eta\sum_{i=1}^{N_{Expert}} g_i^2Proj_{row(A_i)}(\nabla_X\mathcal{L}), \quad (10)$$

in which the derivation step (9) denotes the conventional Riemannian preconditioner scaling. It should be interpreted that (10) consists of an ensemble of projections of the full matrix gradient onto the row spaces of $A$ experts and the column spaces of $B$ experts.

### 3.2. Rescaling Preconditioners

Equation (10) presents a squared-value weighted sum of an ensemble of gradient projections. Generally, more activated experts lead to smaller per-expert gate values and so lead to a more reduced assembled gradient; On the other hand, more balanced experts also lead to a more reduced assembled gradient since the basic inequality theorem $\sum_i x_i^2 >= \frac{(\sum_i x_i)^2}{n} = \frac{1}{n}$ satisfies its equality condition when $x_i$s are equal. As a result, the gradient of the full matrix $X$ will be underestimated due to those squared gate values. From the perspective of manifolds and curvature, we explain that by considering $g_i$ in (8) as a manifold scaler, which reduces the size of $B_iA_i$ so that would probably increase its curvature. However, the conventional Riemannian preconditioner failed to take the manifold scaler $g_i$ into consideration, since it is designed for a single LoRA adapter.

To alleviate this squared issue, we assume a further rescaling step for the Riemannian preconditioners:

$$\nabla_{A_i}\mathcal{L} = \frac{(B_i^T B_i)^{-1}\nabla_{A_i}\mathcal{L}}{g_i}$$
$$\nabla_{B_i}\mathcal{L} = \frac{\nabla_{B_i}\mathcal{L}(A_i A_i^T)^{-1}}{g_i}, \quad (11)$$

which is introduced to replace (6) in the derivation of Equation (9), to eliminate the variable $g_i$ and keeps only a first power of $g_i$ in the final equation (10). Throughout this transformation, the final ensemble of multi-expert projections shares an equivalent scale with the projection of a single LoRA adapter, shown in Equation (12). Therefore, training of an MoE-LoRA will be alleviated from under-estimation.

$$X_{new} = X - \eta \sum_{i=1}^{N_{Expert}} g_i Proj_{col(B_i)}(\nabla_X\mathcal{L})^T$$
$$-\eta \sum_{i=1}^{N_{Expert}} g_i Proj_{row(A_i)}(\nabla_X\mathcal{L}). \quad (12)$$

We claim that (12) further approaches global fully finetuning because of two basic reasons: Firstly, it is derived by implementing Riemannian preconditioners to calibrate each LoRA expert's gradient (given by (6)), thus ensuring each LoRA expert can get close to their respective local full-rank training behavior (i.e., per-expert fully-finetuning equivalency), according to Zhang et al. (Zhang & Pilanci, 2024); Secondly, since there exists a further distortion of each expert space introduced by their respective gate value, leading to an inconsistency between per-expert local optimals and the global optimal. Therefore, (11) further introduces a respective gate value $g_i$ as a re-scaler to each expert's Riemannian preconditioner to relieve the expert distortion resulting from the multiplication of gate value during forwarding. Derived by (11) and (9), (12) is achieved. As a result, (12) can further approach global fully-finetuning equivalency. For example, larger gate values introduce less distortion. Thus, through (12) experts with larger gate values are re-scaled less than those with smaller ones.

### 3.3. Engineering Approximation

Although Equation (11) provides an approach to eliminate under-estimation for MoE-LoRA, it is unrealizable since each LoRA module exists a respective $g_i$ for every single token of every batch sample. Actually, during the training, backpropagation always runs after averaging all the losses of each single token of each sample in a batch. Thus, it is impossible to reconstruct and rescale the respective gradient contributed by each single token when we optimize a LoRA module. Alternatively, we design an engineering approximation to (11) and (12), by replacing each gate value $g_i$ with

its square root $\sqrt{g_i}$ during model forwarding. Consequently, Equation (12) can be achieved only under the preconditioners of (6), because the quadratic terms of gate values $g_i^2$ in Equation (10) are now naturally become linear terms $g_i$.

Replacing $g_i$ by $\sqrt{g_i}$ simultaneously introduces destruction to forwarding, as the sum of square roots does not equal $1$. One possible solution is to re-normalize those square roots to be summed up as $1$. However, it brings inconsistency between the assigned weights of experts during forwarding and backwarding. Therefore, we propose another strategy to accommodate both aspects, which is manually assigning optimizable and unoptimizable components of Equation (8), to satisfy the requirements of both forwarding in (8) and backwarding in (12). During the forwarding process, the proposed strategy is simply expressed by:

$$X = \hat{W} + \sum_{i=1}^{N_{Expert}} \sqrt{g_i}B_i A_i + (g_i - \sqrt{g_i})\hat{B}_i\hat{A}_i, \quad (13)$$

where we define the hat symbol $\hat{\ }$ as an operation of gradient detaching. Specifically, a $\hat{p}$ denotes that the variable $p$ does not require gradient, which also means $p$ should be detached from gradient tracking along the whole neural network. By decomposing optimizable and unoptimizable components like this, low-rank matrices $A$ and $B$ are able to be optimized following (12) (see Appendix E for detailed derivations), while the equivalent behavior and the same result of conventional module forwarding are still preserved. Moreover, by maintaining the optimizable $g_i$ terms in forwarding and treating all the $\sqrt{g_i}$ as constants that are not subject to optimization, (13) also keeps the conventional training behaviors of gates. Additionally, this modification introduces only a minimal overhead to the original forward computation process.

## 4. Experiments

We present a series of comparative experiments to evaluate the performances of MoE-LoRA across various downstream tasks (Zhang et al., 2024b; Luo et al., 2025b) including Question Answering, the GLUE Benchmark, and the Vision-Language task. Specifically, two types of experimental candidates are mainly involved in our experiments: (1) MoE-LoRA with experts updated independently using Riemannian scaled optimizer; and (2) MoE-LoRA updated using Riemannian scaled optimizer, plus incorporating our proposed rescaling technique (the engineering approximation). We implement both of them on SGD and AdamW optimizers respectively. As a further reference, we also exhibit our comparisons and possibility of integrations with previous MoE-LoRA baselines, such as MoLA (Gao et al., 2024). Finally, to lend support to our theoretical foundation, we conduct an ablation study by assessing our forwarding

*Table 1.* Question answering evaluations across four QA datasets with Llama-3.2-3B as the foundation model. Our gate-based rescaling methodology outperforms conventional Riemannian preconditioned optimizers, in terms of both SGD and AdamW. Each pair of comparing candidates is trained through the same steps until they both achieve good stable performances.

| | ScienceQA | CommonsenseQA | OpenBookQA | SIQA | avg. |
|---|---|---|---|---|---|
| $RSGD_{20,10,4}$ | 62.8 | 52.4 | 53.2 | 65.7 | 58.5 |
| $gRSGD_{20,10,4}$ | **70.1** | **55.4** | **59.8** | **68.5** | **63.5** |
| $RAdamW_{20,10,4}$ | 82.6 | 67.7 | 70.4 | 81.5 | 75.6 |
| $gRAdamW_{20,10,4}$ | **83.8** | **68.2** | **72.4** | **82.3** | **76.7** |

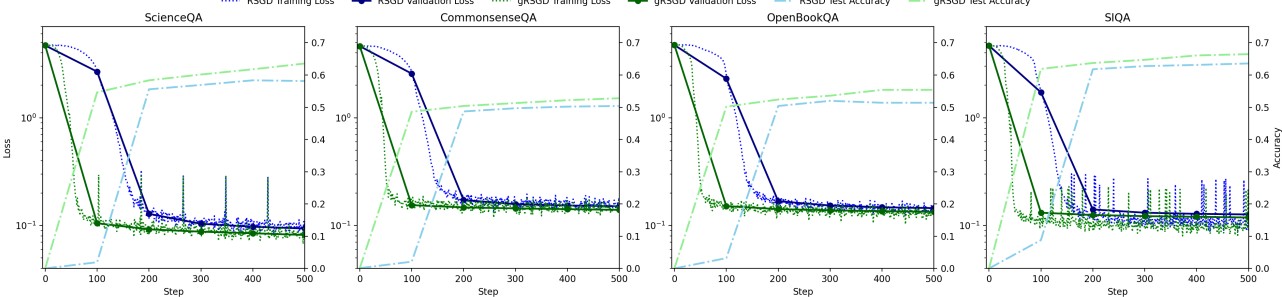

*Figure 2.* Converging Performances of $RSGD_{20,10,4}$ and $gRSGD_{20,10,4}$ MoE-LoRA with Llama-3.2-3B as the foundation model. The x-axis represents training steps, the left y-axis in each figure represents the training or validation losses, while the right y-axis in each figure represents the accuracy metrics of test sets. Before implementing our gate-based rescaling method, the training and validation losses of RSGD optimizer across four tasks are significantly reduced around 100-200 steps, while after implementing our method, they are significantly reduced earlier around 0-100 steps.

revisions only under a classic optimizer without Riemannian preconditioners support.

### 4.1. Experimental Setup

For most experiments, unless otherwise specified, we construct a mixture of LoRAs modules with a total of 20 experts, a rank of 4 for each expert, and a selection of top-10 experts activated each time. Furthermore, a range of other architectural MoE settings are also discussed in the ablation section. We perform experiments based on Llama-3.2-3B (Touvron et al., 2023), GLM-4-9B (Zeng et al., 2024), and LLaVA-v1.5-7B (Liu et al., 2024a) as the foundation models. During training, we follow a linear decay learning-rate scheduler. We assign a relatively smaller learning rate to gate module compared to other trainable components, to achieve a stable training behavior. The reduced learning rate for gate helps to prevent model from experiencing abrupt and erratic routing changes. For further stabilization, we also cap its maximum gradient norm at 1.0. We carefully assign different initial learning rates for various tasks, trying to ensure all models achieve their best performances in a capable running time.

We denote the number of experts, top-k, and the per-expert rank as $n, k, r$ respectively; For experimental candidates using conventional Riemannian preconditioned optimizers, we denote them as $RSGD_{n,k,r}$ and $RAdamW_{n,k,r}$, in which the front $R$ represents the word *Riemannian*; While

those candidates integrated with our gate-based rescaling approach are denoted as $gRSGD_{n,k,r}$ and $gRAdamW_{n,k,r}$ respectively, in which the front $g$ represents that we rescale the gradient by gate values.

### 4.2. Question Answering Evaluations

We evaluate our proposed method on several question-answering benchmarks, including ScienceQA (Lu et al., 2022), CommonsenseQA (Talmor et al., 2019), Open-BookQA (Mihaylov et al., 2018) and SIQA (Sap et al., 2019). These question-answering datasets encompass a diverse range of domains and types, such as science, social interactions, common sense, and open-book exams, etc. We implement all the experimental candidates based on Llama-3.2-3B as their foundation model. For the SGD optimizer, we set an initial learning rate to $3 \times 10^{-5}$ for every LoRA expert; For the AdamW optimizer, we utilize an initial learning rate of $1 \times 10^{-5}$. We run through all the experiments until they are stabilized at a stable performance, and especially ensure that each pair of comparing candidates (i.e., independently Riemannian preconditioned MoE-LoRA, and that with our proposed rescaling approach) are trained through the same steps to make sure they are fairly comparable. Specifically, depending on the complexity of datasets, we choose from two settings, 800 or 1,400 steps, for all the QA evaluations, except that of $RAdamW$ and $gRAdamW$

*Table 2.* GLUE Benchmark evaluations across nine tasks with Llama-3.2-3B and GLM-4-9B as the foundation models. Our gate-based rescaling method contributes an overall improvement over GLUE Benchmark, in terms of Riemannian preconditioned SGD and AdamW.

| | | CoLA | SST-2 | MRPC | STS-B | QQP | MNLI | QNLI | RTE | WNLI | avg. |
|---|---|---|---|---|---|---|---|---|---|---|---|
| Llama 3.2 (3B) | $RSGD_{20,10,4}$ | 48.25 | 93.58 | 73.57 | 48.15 | 80.22 | 62.42 | 78.39 | 62.45 | 39.44 | 65.16 |
| | $gRSGD_{20,10,4}$ | **55.86** | **94.95** | **76.29** | **56.70** | **83.48** | **80.37** | **83.42** | **72.92** | **52.11** | **72.90** |
| | $RAdamW_{20,10,4}$ | 65.94 | 96.56 | 70.03 | 60.12 | 85.36 | 81.36 | **91.55** | 53.95 | 45.07 | 72.22 |
| | $gRAdamW_{20,10,4}$ | **67.38** | **96.67** | **81.57** | **61.08** | **86.35** | **81.59** | 91.32 | **55.75** | **47.89** | **74.40** |
| GLM 4 (9B) | $RSGD_{20,10,4}$ | 62.33 | 91.74 | **82.32** | 63.06 | 85.36 | 81.67 | 90.27 | 62.59 | 76.05 | 77.27 |
| | $gRSGD_{20,10,4}$ | **62.97** | **95.18** | 81.68 | **63.68** | **86.84** | **86.27** | **91.17** | **80.21** | **77.46** | **80.61** |
| | $RAdamW_{20,10,4}$ | **62.54** | 45.53 | 79.30 | 64.15 | 88.72 | 88.03 | 91.58 | 89.56 | 84.51 | 77.10 |
| | $gRAdamW_{20,10,4}$ | 62.25 | **46.22** | **79.59** | **64.36** | **88.82** | **88.19** | **91.66** | **91.37** | **85.92** | **77.60** |

on CommonsenseQA, which we train up to 2,000 steps to achieve a more clear distinction between two comparable candidates. We present our evaluated performances in Table 1. It is observed that: (1) Riemannian preconditioned Optimizers incorporating our approach achieve better performances for every QA benchmark, albeit with varying degrees of improvement; (2) Overall, we exhibit more contribution to Riemannian preconditioned SGD than to that of AdamW: We improve the performance of $RSGD$ by around 8.5%, while we improve $RAdamW$ by around 1.5%.

Besides our improvements in final performances, we also witness a boost in terms of converging speed under our optimization. To clearly display this, we plot loss-decreasing curves and metric variations of the four question-answering datasets under SGD optimizer in Figure 2. It is clearly shown that $gRSGD$ converges faster than $RSGD$, in terms of training and evaluating losses as well as accuracy metrics.

### 4.3. Performance on GLUE Benchmark

To comprehensively examine our effectiveness, we perform a series of downstream evaluations on the benchmark of GLUE (Wang et al., 2019), which is a collection of resources for evaluating model performances on natural language understanding. We first run through all the evaluations in GLUE with Llama-3.2-3B as the foundation model and present the benchmark results in Table 2. For most SGD experiments we set an initial learning rate for LoRA experts as $3 \times 10^{-5}$, except WNLI for which we set its initial learning rate to $3 \times 10^{-6}$; For AdamW experiments we choose an initial learning rate from $\{3 \times 10^{-5}, 1 \times 10^{-5}\}$. For most datasets, we train for 2,000 steps, excluding some AdamW experiments in which we perform an early stop at around 1,000 since they appear to be converged or even overfitting. Table 2 illustrates our effectiveness across various downstream applications as well as the overall assessment under Llama-3.2-3B. In terms of overall performances, our approach improves $RSGD$ and $RAdamW$ by 11.9% and 3.0% respectively.

*Table 3.* Visual7W and VMCBench performances after trained for 1,000 steps, with LLaVA-v1.5-7B as the foundation model. (For VMCBench, we use 100 samples to evaluate, thus the accuracy will be at most a two-digit decimal. That's why we list all numbers in percentage here for a more comfortable present.)

| | Visual7W | VMCBench |
|---|---|---|
| $RSGD_{20,10,4}$ | 68% | 59% |
| $gRSGD_{20,10,4}$ | **72%** | **70%** |
| $RAdamW_{20,10,4}$ | 77% | 75% |
| $gRAdamW_{20,10,4}$ | **78%** | 75% |

Subsequently, we extend experiments to a larger foundation model, GLM-4-9B. Since the 9B model is more powerful in few-shot learning, for some datasets such as SST-2 etc., we set lower learning rates such as $3 \times 10^{-6}$ and $1 \times 10^{-6}$ respectively for SGDs and AdamWs, to make sure a clear loss decreasing period can be witnessed. We train for the same number of steps for each pair of competitive candidates. Table 2 also illustrates the performances of training MoE-LoRA through different optimizing strategies with GLM-4-9B. Results still witness our overall outperformance. In particular, we improve the average performance of $RSGD$ by around 4.3%, and that of $RAdamW$ by around 0.7%.

### 4.4. Performance on LLaVA

Beyond pure textual tasks (Bi et al., 2025), vision-language cross-modal tasks have garnered increasing attention in recent years, witnessing the emergence of notable achievements such as LLaVA, CogVLM, etc. (Chen et al., 2024; Wang et al., 2024c; Ge et al., 2021; Jin et al., 2024) Thus, we further evaluate our gate-based rescaling approach in computer vision (Feng et al., 2022; 2025). Specifically, we implement an MoE-LoRA architecture for the well-known vision-language foundation model, LLaVA-v1.5-7B (Chen et al., 2024). We introduce trainable MoE-LoRA adapters into both visual and textual modules of LLaVA-v1.5-7B. For evaluation, Visual7W (Zhu et al., 2016) and VMCBench (Zhang et al., 2025b) datasets are employed,

which both consist of multimodal samples each containing a multiple-choice question paired with a related image. The question can be answered through understanding the provided image. Visual7W is a subset of Visual Genome (Krishna et al., 2017) dataset, while VMCBench is a benchmark created from 20 existing VQA datasets. For VMCBench, we only use their dev set since their test set is not labeled. We take 900 of all the 1,000 labeled samples as training samples, while the rest 100 are for evaluation. Table 3 exhibits the results of all experimental candidates. Our approach consistently demonstrates visible improvements, especially for SGD.

### 4.5. Compare and Integrate with MoE-LoRA Baselines

We then compare and integrate our method with existing MoE-LoRA baselines. We provide our comparisons with two baselines: (1) The pure mixture of LoRAs (Liu et al., 2023), which we denote as MoELoRA and use token-level routing; (2) MoLA (Gao et al., 2024), which is a MoE-LoRA variant specifically focusing on assigning different numbers of experts to different layers, and proving that higher layers need more LoRA experts. It should be noted that our proposed gate-based rescaling approach can be integrated with most MoE-LoRA variants since they are not in conflict. Take MoLA as an example, we can integrate our method with MoLA by implementing a model with more experts in its higher layers and trained through Riemannian preconditioners and gate-based rescaling approach. We reproduce MoELoRA and MoLA, implement the integrations, and illustrate their performances in Table 4. We use Llama-3.2-3B as the foundation model and follow MoLA's configurations here, which means we set the per-expert rank to 4, top-k to 2, and the total number of experts of all layers to 140. In this way, MoELoRA and our method assign 5 experts to each layer, while MoLA assigns 2, 4, 6, and 8 experts respectively to the bottom, lower middle, higher middle, and top layers. In Table 4 we denote this special assignment strategy as (2,4,6,8), while the average assignment is (5,5,5,5), where each digit covers seven layers under Llama-3.2-3B. We still provide enhancement in the context of MoLA architecture.

### 4.6. Ablation Study

**Theoretical Dependence.** Although our proposed approach is grounded in the context of Riemannian preconditioners, it is important to note that our engineering implementation does not inherently require coexistence with Riemannian preconditioners. The reason is that our modifications are solely focused on altering the forward propagation conventions of MoE-LoRA. This consequently raises a vital question about the standalone efficacy of our modifications in enhancing MoE-LoRA's performance, without depending on the Riemannian preconditioning context. Ideally, since the conventional un-preconditioned optimizer does not guar-

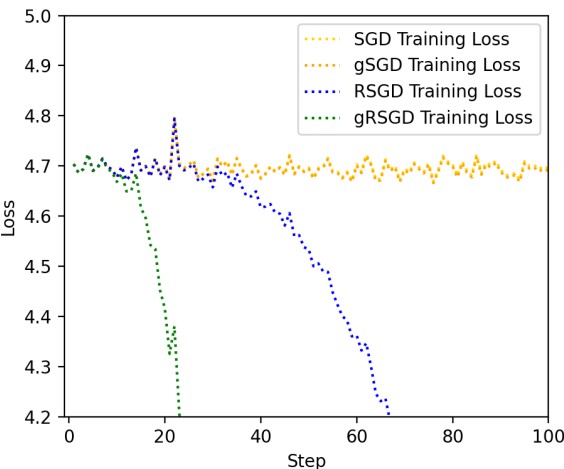

*Figure 3.* Curves of ScienceQA training losses under the optimization of conventional and Riemannian preconditioned SGDs, and also both integrated with the gate-based rescaling approach. Llama-3.2-3B serves as the foundation model.

antee a projection of full matrix gradient in low-rank space, it should be trivial for them to normalize the sum of expert gradients by replacing $g_i$ with $\sqrt{g_i}$. To confirm this, we conduct an ablation study by integrating our gate-based revision with a conventional un-preconditioned SGD optimizer. The loss-decreasing curves shown in Figure 3 illustrate that applying our approach directly on a pure SGD optimizer does not provide help, which oppositely demonstrates our refinement is highly coupled with the Riemannian preconditioning algorithm.

**Various MoE architectures.** To demonstrate that our proposed approach can be generalized to various settings of LoRA mixtures, we construct different MoE-LoRA architectures for further exploration, including the variations in the numbers of experts, per-expert ranks, and the number of top-k. Specifically, we test seven structural conditions on the ScienceQA dataset, and all candidates are trained for 800 steps using the same initial learning rate. Table 5 exhibits the results, showing we are able to outperform across most circumstances in terms of MoE structure. Moreover, it is also observed from SGD performances that, variations in expert numbers or per-expert ranks introduce limited impacts on our effectiveness, while larger top-k roughly exhibit higher boosts. This observation aligns with our theoretical analysis, which suggests a larger number of activated experts results in more reduced per-expert gate values, thereby leaving a larger margin for our revision to take effect.

## 5. Conclusion

We introduce the Riemannian gradient preconditioners to train a mixture of Low-rank Experts (MoE-LoRA). Instead

*Table 4.* Baselines Comparison and Integration. The first three lines provide comparisons between pure MoE-LoRA (MoELoRA-$SGD$), MoLA (MoLA-$SGD$), and our gate-rescaled Riemannian preconditioning method (MoELoRA-$gRSGD$). The last two lines provide MoLA integrated with conventional (MoLA-$RSGD$) and gate-rescaled preconditioning methods (MoLA-$gRSGD$), respectively. All candidates are trained using SGD optimizers for up to 2,000 steps.

|  | Experts | ScienceQA | CommonsenseQA | OpenBookQA | SIQA | avg. |
|---|---|---|---|---|---|---|
| MoELoRA-$SGD$ | (5,5,5,5) | 54.68 | 48.90 | 48.40 | 57.92 | 52.47 |
| MoELoRA-$gRSGD$ (Ours) | (5,5,5,5) | 70.01 | 54.80 | 63.60 | 64.45 | 63.22 |
| MoLA-$SGD$ | (2,4,6,8) | 54.99 | 49.20 | 52.80 | 58.94 | 53.98 |
| MoLA-$RSGD$ | (2,4,6,8) | 68.03 | 53.90 | 59.80 | 64.08 | 61.45 |
| MoLA-$gRSGD$ (Ours) | (2,4,6,8) | **70.46** | **56.30** | **64.00** | **64.90** | **63.92** |

*Table 5.* Accuracies and boosts of ScienceQA for conventional and gate-rescaled Riemannian optimizers under various MoE architectures. Llama-3.2-3B serves as the foundation model.

| $n/k/r$ | $RSGD$ | $gRSGD$ | Boost | $RAdamW$ | $gRAdamW$ | Boost |
|---|---|---|---|---|---|---|
| 5/5/4 | 65.78 | **71.31** | 8.41%↑ | 76.67 | **78.19** | 1.98%↑ |
| 8/5/4 | 65.11 | **69.96** | 7.45%↑ | 78.69 | **79.45** | 0.97%↑ |
| 10/5/4 | 64.34 | **69.33** | 7.76%↑ | 79.05 | **80.31** | 1.59%↑ |
| 10/5/2 | 72.03 | **78.06** | 8.37%↑ | 79.63 | **80.13** | 0.63%↑ |
| 10/5/1 | 79.95 | **87.28** | 9.17%↑ | 80.71 | **81.11** | 0.50%↑ |
| 10/10/2 | 68.62 | **77.16** | 12.45%↑ | 77.47 | **77.65** | 0.23%↑ |
| 10/2/2 | 79.68 | **81.16** | 1.86%↑ | **83.63** | 83.45 | -0.22%↓ |

of directly attaching Riemannian preconditioners to each expert's gradient for pursuing local optimality, we claim that multiplying expert $B_iA_i$ by its respective gate value $g_i$ during forwarding leads to a further rescaling of the manifold constructed by expert $i$. To alleviate this, Riemannian preconditioners designed for MoE-LoRA shall be revised to incorporate gate values. To approximate this concept, we propose an engineering solution that decomposes forwarding variables into optimizable and un-optimizable components. Experiments across various downstream tasks demonstrate our performance improvement over conventional Riemannian preconditioners. Ablation studies further demonstrate our theoretical foundation and universality. We claim that, our work can be applied to the fields like efficient and low-resource model training, continual or multi-task learning, stablized training and modular task adaptation, etc.

## Acknowledgments

This work has been supported by the National Natural Science Foundation of China (NSFC) for Distinguished Young Scholar under Grant 62425601, the NSFC under Grant 62406036, the National Key Research and Development Program of China under Grant 2024YFC3308500, the State Key Laboratory of Networking and Switching Technology under Grant NST20250110, the New Cornerstone Science Foundation through the XPLORER PRIZE, and also sponsored by SMP-Zhipu.AI Large Model Cross-Disciplinary Fund under Grant ZPCG20241029322.

## Impact Statement

This paper presents work whose goal is to advance the field of Machine Learning. There are many potential societal consequences of our work, none of which we feel must be specifically highlighted here.

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

# A. Various Model Sizes and MoE Architectures

To evaluate our method, we already conducted experiments on Llama-3.2-3B, GLM-4-9B, and LLaVA-v1.5-7B. We also presented our $n/k/r$ analysis in Table 5 in Section 4.6, which consists of seven different candidates tested under SGD and AdamW optimizers with Llama-3.2-3B serving as their foundation model. To make our investigation further sufficient, here we add Llama-3.2-1B as a new foundation model, and add two new experiments under LLaVA-v1.5-7B with two different $n/k/r$ configurations (16/8/4 and 10/5/4).

Specifically, we conduct further experiments on Llama-3.2-1B for four QA benchmarks, each trained for 2,000 steps. To speed up the training, we set a relatively smaller MoE configuration which is $10/5/1$. Results are illustrated in Table 6. For those two new $n/k/r$ configurations of LLaVA-v1.5-7B, we conduct their experiments on both Visual7W and VMCBench benchmarks under SGD and AdamW optimizers. An overall enhancement of our method can still be witnessed under different configurations (especially for SGD), illustrated by Table 7.

*Table 6.* Question answering evaluations across four QA datasets with Llama-3.2-1B as the foundation model.

| | ScienceQA | CommonsenseQA | OpenBookQA | SIQA | avg. |
|---|---|---|---|---|---|
| $RSGD_{10,5,1}$ | 47.71 | 49.47 | 48.80 | 50.41 | 49.10 |
| $gRSGD_{10,5,1}$ | **49.87** | **59.30** | **54.00** | **57.06** | **55.06** |
| $RAdamW_{10,5,1}$ | 46.18 | 42.92 | 41.60 | 44.11 | 43.70 |
| $gRAdamW_{10,5,1}$ | **46.58** | **43.82** | **43.40** | **45.50** | **44.83** |

*Table 7.* Visual7W and VMCBench Performances of LLaVA-v1.5-7B across various MoE architectures.

| $n/k/r$ | Task | $RSGD$ | $gRSGD$ | $RAdamW$ | $gRAdamW$ |
|---|---|---|---|---|---|
| 10/5/4 | **Visual7W** | 71% | **74%** | 76% | 76% |
| | **VMCBench** | 63% | **73%** | 76% | **77%** |
| 16/8/4 | **Visual7W** | 72% | **74%** | 76% | **77%** |
| | **VMCBench** | 59% | **69%** | 71% | 71% |

# B. Convergence Efficiency

In the main body of our paper, we illustrate the converging speed enhancements of our proposed approach, $gRSGD$, over the conventional $RSGD$ through a series of loss-decreasing plots. To further exhibit the comprehensive comparisons of convergence efficiency, we provide more results on GLUE benchmarks. In particular, for experiments conducted under Llama-3.2-3B as well as GLM-4-9B, we record metrics after the initial 100 training steps for each of the GLUE evaluations, as detailed in Table 8 and Table 9 respectively.

Table 8 and 9 clearly demonstrate the superior convergence speed of our solutions over the conventional Riemannian preconditioned SGD optimizers. Nevertheless, they simultaneously illustrate an overall equivalent performance with trivial differences between our gate-based approach and the conventional Riemannian preconditioning method under AdamW optimizers. This indicates that our proposed approach is more valuable for SGD optimization. AdamW optimizers already present robust converging performances due to their adaptive gradient and learning rate mechanisms. As a result, our global optimal approximation under AdamW optimizing mainly contributes to the final optimality rather than significantly accelerating the initial gradient descending.

*Table 8.* GLUE Benchmark evaluations after the initial 100 training steps conducted under Llama-3.2-3B. Our proposed gate-based rescaling method contributes an overall converging speed enhancement over conventional Riemannian preconditioned SGD optimizers, while for AdamW optimizers, we provide a similar converging speed compared with the conventional Riemannian ones.

| | CoLA | SST-2 | MRPC | STS-B | QQP | MNLI | QNLI | RTE | WNLI | avg. |
|---|---|---|---|---|---|---|---|---|---|---|
| $RSGD_{20,10,4}$ | 22.52 | 91.63 | 49.97 | 0.00 | 62.22 | 36.27 | 58.90 | 56.68 | 1.41 | 42.18 |
| $gRSGD_{20,10,4}$ | **39.30** | **92.66** | **66.55** | **24.49** | **66.17** | **43.40** | **65.33** | **63.18** | **26.76** | **54.20** |
| $RAdamW_{20,10,4}$ | **52.05** | 92.66 | **68.99** | 54.04 | **72.11** | 60.81 | **84.14** | 53.95 | 32.39 | 63.46 |
| $gRAdamW_{20,10,4}$ | 50.63 | **93.23** | 67.01 | **54.72** | 71.41 | **61.96** | 83.80 | **55.75** | **40.85** | **64.37** |

*Table 9.* GLUE Benchmark evaluations after the initial 100 training steps conducted under GLM-4-9B. Our proposed gate-based rescaling method still contributes an overall converging speed enhancement over conventional Riemannian preconditioned SGD optimizers, while for AdamW optimizers, we still provide a similar converging speed compared with the conventional Riemannian ones. (Note that for CoLA, SST-2, and MRPC, we utilize a lower initial learning rate such as $3 \times 10^{-6}$ and $1 \times 10^{-6}$, while the others are $3 \times 10^{-5}$ and $1 \times 10^{-5}$. Therefore CoLA, SST-2, and MRPC converge much slower than the others.)

| | CoLA | SST-2 | MRPC | STS-B | QQP | MNLI | QNLI | RTE | WNLI | avg. |
|---|---|---|---|---|---|---|---|---|---|---|
| $RSGD_{20,10,4}$ | 0.00 | 0.00 | 0.41 | 61.15 | 84.57 | 62.35 | 87.49 | 41.73 | 63.38 | 44.56 |
| $gRSGD_{20,10,4}$ | **7.22** | **48.97** | **62.26** | **63.06** | **85.95** | **85.66** | **89.28** | **75.18** | **74.65** | **65.80** |
| $RAdamW_{20,10,4}$ | **17.37** | 0.00 | 0.00 | 63.34 | **86.15** | 0.00 | 89.05 | 89.21 | **80.28** | 47.27 |
| $gRAdamW_{20,10,4}$ | 16.75 | 0.00 | 0.00 | **63.54** | 86.05 | **0.61** | **89.34** | **91.37** | 78.87 | **47.39** |

## C. AdamW Weight Decay Analysis

AdamW implements a strategy called weight decay, which decays the trainable weights after each gradient update by $\theta_t = \theta_t - \alpha\lambda\theta_t$. Instead of the original Adam algorithm, AdamW separates the weight decay from the gradient update, which leads to better performance in some cases. To comprehensively prove the effectiveness of our gate-based rescaling method over Riemannian preconditioned AdamW, we evaluate our boosts across various weight decay factors $\lambda$. Results are exhibited in Table 10.

*Table 10.* ScienceQA boosting performances under Llama-3.2-3B, across different AdamW weight decay.

| | 0 | 1e-5 | 1e-4 | 1e-3 |
|---|---|---|---|---|
| $RAdamW_{20,10,4}$ | 82.60 | 83.50 | 83.23 | 83.99 |
| $gRAdamW_{20,10,4}$ | **83.80** | **84.58** | **84.98** | **84.67** |
| **Boost** | 1.45%↑ | 1.30%↑ | 2.10%↑ | 0.81%↑ |

## D. Multi-Task Performance

One of the most valuable features of MoE architectures is their capability of modeling multiple tasks. Through gating mechanism, the MoE system adeptly delegates specific tasks to individual experts, thereby facilitating a more focused and efficient learning process within each expert module. As a result, one question arises regarding our proposed gate-based rescaling approach: Can it still effectively augment the performance of MoE architectures in multi-task scenarios?

To illustrate this, we manually construct two mixed datasets, each consisting of three irrelevant natural language tasks from the GLUE Benchmark. The first mixture consists of CoLA, SST-2 and MRPC tasks, which serves as a multi-task scenario involving both grammar checking, sentiment classification, and equivalent sentences judging; The second mixture consists of STS-B, QQP and QNLI tasks, which serves as another multi-task scenario involving both sentence similarity scoring, equivalent questions judging, and question-answering NLI. For evaluation, we test candidates on each of the tasks individually and then average per-task performances within the mixture as the overall evaluation for that mixture.

For sufficiently assessing the multi-task performance of our proposed gate-based rescaling method, we conduct experiments under two different MoE configurations, i.e., $20/10/4$ and $10/5/4$. We train each candidate for 2,000 steps under $RAdamW$ and $gRAdamW$ ($RAdamW$ with our proposed gate-based rescaling method) optimizers. Results are exhibited in Table 11. Our proposed method is still effective to boost feature learning under multi-task scenarios

*Table 11.* Conventional and gate-rescaled optimizers performed on two mixed datasets consisting of tasks from the GLUE Benchmark. All candidates are trained for 2,000 steps. Our gate-based rescaling method still contributes enhancements.

| **Mixture** | $n/k/r$ | $RAdamW$ | $gRAdamW$ |
|---|---|---|---|
| **CoLA + SST-2 + MRPC** | 20/10/4 | 70.15 | **71.39** |
| | 10/5/4 | 71.64 | **72.13** |
| **STS-B + QQP + QNLI** | 20/10/4 | 74.61 | **75.74** |
| | 10/5/4 | 74.81 | **75.36** |

## E. Backward Derivation of the Engineering Strategy

We further elaborate why we implement (13), which is our engineering approximation for achieving (11) and (12). Since by forwarding as (13), the gradient updating process of $X$ can be derived as the following (Similar as previous, we treat gate value $g_i$s as constants when focusing on the gradients of $A_i$s and $B_i$s):

$$
\begin{aligned}
X_{new} =& \hat{W} + \sum_{i=1}^{N_{Expert}} [\sqrt{\hat{g_i}}(B_i - \eta\nabla_{B_i}\mathcal{L})(A_i - \eta\nabla_{A_i}\mathcal{L}) + (g_i - \sqrt{\hat{g_i}})\hat{B_i}\hat{A_i}] \\
=& (\hat{W} + \sum_{i=1}^{N_{Expert}} \sqrt{\hat{g_i}}B_iA_i + (g_i - \sqrt{\hat{g_i}})\hat{B_i}\hat{A_i}) - \eta \sum_{i=1}^{N_{Expert}} \sqrt{\hat{g_i}}[B_i(\nabla_{A_i}\mathcal{L}) + (\nabla_{B_i}\mathcal{L})A_i] \\
=& X - \eta \sum_{i=1}^{N_{Expert}} \sqrt{\hat{g_i}}(B_i\nabla_{A_i}\mathcal{L} + \nabla_{B_i}\mathcal{L}A_i) \\
=& X - \eta \sum_{i=1}^{N_{Expert}} (\sqrt{\hat{g_i}})^2 Proj_{col(B_i)}(\nabla_X\mathcal{L})^T - \eta \sum_{i=1}^{N_{Expert}} (\sqrt{\hat{g_i}})^2 Proj_{row(A_i)}(\nabla_X\mathcal{L}) \quad \text{(same as the derivation of (10))} \\
=& X - \eta \sum_{i=1}^{N_{Expert}} g_i Proj_{col(B_i)}(\nabla_X\mathcal{L})^T - \eta \sum_{i=1}^{N_{Expert}} g_i Proj_{row(A_i)}(\nabla_X\mathcal{L}),
\end{aligned}
$$

so that (12) can be achieved.

The advantages of implementing (13) can be elaborated from two aspects: Firstly, it achieves (12) when conducting gradient updating of $X$ while still keeps the original behavior of training gates, because it holds the same gate gradient, $\nabla_{g_i}X = A_i^T B_i^T$, as normal forwarding; Secondly, it provides equivalent behavior and the same result of normal module forwarding $X = \hat{W} + \sum_{i=1}^{N_{Expert}} g_iB_iA_i$, and only requires a relatively low overhead.

## F. Method Implementation

The engineering alternative solution of the gate-based rescaling approach is to manually separate the forwarding into optimizable and unoptimizable components. Here we provide our implementation in Python-like pseudocode. We only update two lines of the original MoE-LoRA code.

---

**Algorithm 1** Engineering Alternative Solution of Gate-based Rescaling Method

```
def forward(self, x, ...):
    ...
    # compute gate values
    gvs = ...
    ...
    # execute each activated expert
    for exp_id in activated_experts:
        A = self.As[exp_id]
        B = self.Bs[exp_id]
        gv = gvs[:,:,exp_id]
        exp_out = B(A(x))
        sqrt_gv = (gv**0.5).detach() # update 1
        w_exp_out = sqrt_gv*exp_out+(gv-sqrt_gv)*exp_out.detach() # update 2
        result = result + w_exp_out
        ...
```

---

## G. Experimental Details

We present our experimental details in Table 12. All experiments in this paper follow this configuration unless they specify their particular settings. For training steps, some of the experiments may converge earlier, therefore we perform an early

stop for those experiments. We constrain the maximum of training steps by 2,000, considering it a relatively fair setup for various downstream tasks, especially those with different scales of training corpora but in the same level of complexity.

*Table 12.* Default experimental details implemented throughout this paper. All experiments follow this configuration unless they specify their particular settings, like the MoE structural experiments, baselines comparing experiments, and the experiments of AdamW weight decay.

| | SGD | AdamW |
|---|---|---|
| Train batch size (logical) | 80 for textual tasks, 40 for vision-language tasks | |
| Max training steps | $<= 2,000$ | |
| Initial lr for 3B (expert) | QA, GLUE: 3e-5
WNLI: 3e-6
Multi-Task: 3e-4 | QA: 1e-5
MRPC, CoLA, QNLI, STS-B: 3e-5
SST-2, QQP, MNLI, WNLI: 1e-5
RTE: 1e-6
Multi-Task: 1e-4 |
| Initial lr for 9B (expert) | CoLA, SST-2, MRPC: 3e-6
STS-B, QQP, MNLI, QNLI, RTE, WNLI: 3e-5 | CoLA, SST-2, MRPC: 1e-6
STS-B, QQP, QNLI, RTE, WNLI: 1e-5
MNLI: 3e-6 |
| Initial lr for 7B (expert) | 3e-5 | 1e-5 |
| Initial lr (gate) | 3e-8 | |
| Lr scheduler (expert) | Linear | |
| Warmup steps | 0 | |
| Max gradient norm (gate) | 1.0 | |
| Default LoRA expert rank | 4 | |
| Default number of experts | 20 | |
| Default activated Top-K | 10 | |
| LoRA $\alpha$ | 16 | |
| LoRA dropout | 0.05 | |
| Default weight decay | / | 0 |
| $\beta_1, \beta_2, \epsilon$ | / | 0.9, 0.999, 1e-6 |

