# OpenReview forum: "A Stronger Mixture of Low-Rank Experts for Fine-Tuning Foundation Models"
_ICML.cc/2025/Conference — ICML 2025 poster_

### Official Review · Reviewer_cEEk · 2025-03-02

**Overall Recommendation:** 4

**Summary:**

The authors propose to apply the Riemannian Preconditioner introduced in the previous work to improve the Mixture of LoRA framework. The Riemannian Preconditioner enhances LoRA training by projecting the full matrix gradient to the subspace of LoRA matrixes, which better approximates full fine-tuning compared to the unscaled gradient descent. However, applying the preconditioner with Mixture of LoRA is coupled with a further rescaling of the manifold constructed for each expert, leading to underestimated gradients. The authors incorporate a new scaling mechanism and develop an engineering approximation to address the issue. Extensive experiments across various downstream tasks including Question Answering, the GLUE Benchmark, and the Vision-Language task are conducted to validate the approach's efficacy.

**Claims And Evidence:**

The overall claims are clearly stated and well supported by extensive experiments across different benchmarks:
a) The authors claim that incorporating a Riemannian Preconditioner into the Mixture of LoRA framework yields superior performance.
b) They claim that the scaling mechanism introduced in the preconditioner helps address the issue of underestimated gradients.
c) They claim that the engineering approximation solution improves training dynamics and model performance.

**Essential References Not Discussed:**

The references cover the related works about Mixture of LoRA and the gradient preconditioners. Works about LoRA and LoRA variants are also covered.

**Experimental Designs Or Analyses:**

The experimental methodology is valid. The authors evaluate their approach across different tasks, from language to vision understanding. I did not detect any significant issues with their design or analysis.

**Methods And Evaluation Criteria:**

Yes. The proposed method builds upon the Riemannian Preconditioner approach from prior work to develop modifications aligned with the Mixture of the LoRA framework. Extensive experiments across various downstream tasks including Question Answering, the GLUE Benchmark, and the Vision-Language task are conducted to validate the approach's efficacy.

**Other Comments Or Suggestions:**

None.

**Other Strengths And Weaknesses:**

This paper combines Mixture of LoRA and Riemannian preconditioning with a rescaling mechanism to address gradient underestimation. While this integration is useful and supported by thorough experiments, the approach remains incremental.The theoretical justification for rescaling appears partially heuristic. Overall, the work provides a moderate advance that could benefit practitioners interested in more effective low-rank fine-tuning.

**Questions For Authors:**

None.

**Relation To Broader Scientific Literature:**

This work combines two established research directions: Mixture of LoRA (Low-Rank Adaptation) and Riemannian Preconditioning. Mixture of LoRA extends the low-rank fine-tuning paradigm by introducing multiple “expert” components. These experts are activated selectively for each token via the gating mechanism. Riemannian Preconditioners are proposed to ensure the update is done in accordance with the full rank gradient projection onto the subspace of LoRA matrices, stabilizing the training process. By merging these ideas and introducing a rescaling mechanism to address gradient underestimation, the paper demonstrates improved performance on various benchmarks. This approach introduces moderate innovation supported by extensive experiment results.

**Theoretical Claims:**

The theoretical claims in this paper is generally correct and sound. The Riemannian Preconditioner is built upon the previous work. They formally derived the preconditioner's form under the Mixture of LoRA scenario. They further propose a rescaling mechanism to address the underestimation issue—although this component lacks a fully rigorous theoretical proof, it is validated by experimental results, which demonstrate improved performance.

---

> ### Author Rebuttal · Authors · 2025-04-01
>
> We appreciate your reviews and thank you for **acknowledging our efforts on theoretical and experimental analysis**. For your concerns mentioned in the review, we provide corresponding response below:
>
> **Response to your concern** of AdamW performances under Multi-Task scenarios
>
> Our supplementary material does not indicate an out-performance of our method for AdamW optimizer under multi-task scenarios. We considered it may due to lacking a sufficient exploring across different multi-task scenarios, since we only conducted the multi-task experiments under a single mixture of two tasks (ScienceQA and MRPC). Consequently, we conducted more experiments on multi-task scenarios in our revision, including two mixtures of tasks and two various configurations of MoE structure.
>
> In our revision, we grouped six tasks from the GLUE Benchmark into two mixtures. The first mixture consists of CoLA, SST-2 and MRPC tasks, which serves as a multi-task scenario involving both grammar checking, sentiment classification, and equivalent sentences judging; The second mixture consists of STS-B, QQP and QNLI tasks, which serves as another multi-task scenario involving both sentence similarity scoring, equivalent questions judging, and question-answering NLI. For evaluation, we tested candidates on each of the tasks individually and then averaged per-task performances within the mixture as the overall evaluation for that mixture.
>
> For sufficiently assessing the multi-task performance of our proposed gate-based rescaling method, we conducted experiments under two different MoE configurations, i.e., $20/10/4$ and $10/5/4$. We trained each candidate for 2000 steps under RAdamW and gRAdamW (RAdamW with our proposed gate-based rescaling method) optimizers. The following two tables illustrate our performances under the first and the second mixtures respectively.
>
> **Mixture 1: CoLA + SST-2 + MRPC**
> |  **Configuration**  |  **$RAdamW$**  |  **$gRAdamW$**  |
> |  -----------------  |  ------------  |  -------------  |
> |      $20/10/4$      |      70.15     |    **71.39**    |
> |      $10/5/4$       |      71.64     |    **72.13**    |
>
> **Mixture 2: STS-B + QQP + QNLI**
> |  **Configuration**  |  **$RAdamW$**  |  **$gRAdamW$**  |
> |  -----------------  |  ------------  |  -------------  |
> |      $20/10/4$      |      74.61     |    **75.74**    |
> |      $10/5/4$       |      74.81     |    **75.36**    |
>
> As a result, we concluded that overall our proposed method is still effective to boost AdamW optimization under multi-task scenarios. We have revised our supplementary material to integrate these experiments and conclusions.

---

### Official Review · Reviewer_1wAK · 2025-03-05

**Overall Recommendation:** 4

**Summary:**

This paper introduces a new approach to enhance the performance of MoE-LoRA for fine-tuning foundation models by incorporating Riemannian Preconditioners. This approach ensures that the gradient updates align more closely with the full-rank optimization, thereby stabilizing and accelerating the training process. Moreover, they identify a previously overlooked issue: the gate values in MoE-LoRA introduce additional scaling that distorts the gradient updates and undermines the effectiveness of Riemannian Preconditioners. To resolve this, the authors propose a novel gate-value-based rescaling method that adjusts the gradients of each expert to account for the impact of gate values. The results show substantial improvements in performance.

**Claims And Evidence:**

Yes, the claims made in the submission are supported by clear and convincing evidence.

**Essential References Not Discussed:**

N/A

**Experimental Designs Or Analyses:**

Yes, I checked the soundness and validity of the experimental designs and analyses presented in the submission. There are no issues with the experimental design.

**Methods And Evaluation Criteria:**

Yes, the proposed methods and evaluation criteria, including the use of benchmark datasets, are appropriate and well-suited for the problem or application at hand.

**Other Comments Or Suggestions:**

Please see weaknesses above, and I recommend that the authors more prominently highlight the contributions of this work in the abstract, introduction, and conclusion.

**Other Strengths And Weaknesses:**

**Strengths are as follows,**
1-This manuscript is written with a theoretical style.

2-This manuscript features comprehensive and in-depth related work, such as conducting theoretical analyses of the most relevant work.

3- This manuscript presents experiments conducted on both QA datasets, the GLUE benchmark, and Multimodal benchmarks. The results consistently show performance improvements compared to RAdamW and RSGD.

4-The experimental results demonstrate that the authors have effectively addressed the two limitations they declared, show that the lift of the method is remarkable.

**Weaknesses are as follows,**
1-They effectively showcased the scalability of their method on Multimodal Large Model LLaVA, which speaks to its broader utility. However, I noticed that they only tested a single configuration of expert numbers. This limits the contribution of this section compared to the rest of the manuscript.

2- Despite testing the performance on multiple foundation models, I am curious to see how the method performs on LLaMA models of varying sizes.

3- In the MoE, the sum of all gating values (g) is constrained to 1. In my experience，I’m not entirely clear on the rationale behind this choice, and it would benefit from a clearer explanation.

4- While the authors have done a thorough job reviewing related work, they should more clearly to highlight the contributions.

5- In Subsection 4.5, it should be “Table 4” instead of “table 4.”

**Questions For Authors:**

Please see the weaknesses 1 and 3.

**Relation To Broader Scientific Literature:**

The manuscript provides a thorough discussion of the relevant literature.

**Theoretical Claims:**

Yes, I checked the correctness of the derivation for the theoretical claims.

---

> ### Author Rebuttal · Authors · 2025-04-01
>
> Thanks for your valuable reviews and your **agreements on our proposed method and our efforts on literature review**. We have conducted several new experiments and provided responses to all your concerns:
>
> **Response to W1** about the limitation of LLaVA experiments
>
> In our revision, we conducted more LLaVA experiments on both Visual7W and VMCBench using LLaVA-v1.5-7B. Specifically, we implemented candidates under two new MoE configurations with different expert numbers: $16/8/4$ and $10/5/4$. The following table illustrates the results. An overall enhancement of our method can still be witnessed under different configurations, especially for SGD.
>
> |  **Candidates**  |   **Visual7W**   |   **VMCBench**  |
> |  :-----   |   -----:  |   -----:  |
> | $RSGD_{16,8,4}$ |     0.72     |    0.59    |
> |$gRSGD_{16,8,4}$ |   **0.74**   |  **0.69**  |
> | $RSGD_{10,5,4}$ |     0.71     |    0.63    |
> |$gRSGD_{10,5,4}$ |   **0.74**   |  **0.73**  |
> | $RAdamW_{16,8,4}$ |   0.76     |   0.71   |
> |$gRAdamW_{16,8,4}$ |  **0.77**  |   0.71   |
> | $RAdamW_{10,5,4}$ |   0.76   |   0.76   |
> |$gRAdamW_{10,5,4}$ |   0.76   | **0.77** |
>
> Our performance boosts for LLaVA under AdamW might not be that remarkable. Therefore, to conduct a further significance analysis of our AdamW boosting, we also implemented more candidates and trained them by AdamW. Please refer to the following table. Similar phenomenon can be invariably witnessed.
>
> |  **Candidates**  |   **Visual7W**   |   **VMCBench**  |
> |  :------   |   ------:  |   ------:  |
> | $RAdamW_{5,5,4}$   |    0.77    |    0.75    |
> | $gRAdamW_{5,5,4}$  |    0.77    |  **0.76**  |
> | $RAdamW_{5,2,4}$   |    0.73    |    0.75    |
> | $gRAdamW_{5,2,4}$  |  **0.77**  |  **0.78**  |
> | $RAdamW_{3,2,4}$   |    0.75    |    0.75    |
> | $gRAdamW_{3,2,4}$  |  **0.76**  |    0.75    |
>
> **Response to W2** about performances on different LLaMA models
>
> We conducted more experiments on LLaMA models besides Llama-3.2-3B. Specifically, among all the LLaMA 3.2 models there are only 1B and 3B models that are purely textual. Therefore, we decided to include further experiments on Llama-3.2-1B. Four QA benchmarks have been tested, each trained for 2000 steps. Due to the limited resources and time, we set a relatively smaller MoE configuration to speed up training, which is $10/5/1$. Results are illustrated in the following table. We also included this table in the revision of our paper.
>
> |  **Candidates**  |  **ScienceQA**  |  **CommonsenseQA**  |  **OpenBookQA**  |  **SIQA**  |  **Avg.**  |
> |  :-----  |  -----:  |  ----:  |  ----:  |  -----:  |  -----:  |
> |  $RSGD_{10,5,1}$     |    47.71    |    49.47    |    48.80    |    50.41    |    49.10    |
> |  $gRSGD_{10,5,1}$    |  **49.87**  |  **59.30**  |  **54.00**  |  **57.06**  |  **55.06**  |
> |  $RAdamW_{10,5,1}$   |    46.18    |    42.92    |    41.60    |    44.11    |    43.70    |
> |  $gRAdamW_{10,5,1}$  |  **46.58**  |  **43.82**  |  **43.40**  |  **45.50**  |  **44.83**  |
>
> Besides LLaMA 3.2 models, we also tested Llama-3.1-8B. Until now we only conducted four ScienceQA evaluations with each training for only 600-800 steps. Results are below:
>
> |  **Candidates**  |  $RSGD$  |  $gRSGD$  |  $RAdamW$  |  $gRAdamW$  |
> |  :-------------  |  -----:  |  ------:  |  -------:  |  --------:  |
> |   **ScienceQA**  |  71.49   |  **76.35**  |  87.50  |  **87.68**  |
>
> **Response to W3** about the sum-to-1 constraint in MoE
>
> In general MoE, constraining the sum of all gate values to 1 contributes to the model stability and probabilistic interpretation. Firstly, it normalizes gate outputs to get rid of some uncontrolled issues during training and inferring, such as gradient explosion or vanishing, overflow of variables from accumulated forwarding, etc. During training, by fixing the sum to 1, the gating network focuses solely on allocating relative importance among experts, rather than learning absolute weight magnitudes. This also reduces the complexity of the optimization problem and stabilizes training; Secondly, sum-to-1 constraint can be interpreted as the probabilistic distribution of selecting each expert, aligning with the requirements of probabilistic models and enabling clearer theoretical foundation of MoE.
>
> **Response to W4 and your suggestion** of prominently highlighting our contributions
>
> Thank you for your suggestions. Our contributions include integrating mixture of LoRAs with Riemannian preconditioners to alleviate both limited representation and sub-optimality issues; emphasizing the distortion issue behind per-expert preconditioning; and proposing a gate-based rescaling method and its engineering approximation to further boost MoE-LoRA training. We have already revised our abstract, introduction and conclusion sections, to explicitly involve these statements.
>
> **Response to W5** about small errors
>
> We revised the mentioned "table 4" to "Table 4" in Section 4.5 in our revision, together with checking and fixing some other small notation errors.

---

> > ### Comment · Reviewer_1wAK · 2025-04-02
> >
> > I thank the authors for their detailed feedback.
> > My main concerns focus on: (i) The underlying principles of the gating values. The authors provide a detailed explanation, which makes it easier for me to understand their core contribution. (ii) The evaluation on LLaVA and LLaMA, which further strengthens their technical contribution. Moreover, I have carefully read the comments from other reviewers and agree with the theoretical contribution and general value of this work. As an additional suggestion, I also recommend that the authors release their code to the community.
> >
> > Overall, the authors addressed all my concerns, and no further issues need to be resolved. I am inclined to accept this work and therefore raised my initial score.

---

> > > ### Author Response · Authors · 2025-04-03
> > >
> > > We really appreciate for your acknowledgements and raising your evaluation of our work. We will release our code to the community after this paper is published. Thank you very much.

---

### Official Review · Reviewer_utbG · 2025-03-07

**Overall Recommendation:** 4

**Summary:**

This paper introduces a training strategy for Mixture-of-Experts (MoE) models with LoRA. It uses Riemannian preconditioning and gate-value scaling to address gradient sub-optimality and representation limitations. The proposed method modifies traditional preconditioners to stabilize gradient updates and improve training robustness. Experiments on NLP and VQA tasks, including QA datasets, the GLUE benchmark, and VG/VMCBench, demonstrate faster convergence and enhanced performance.

## update after rebuttal
The authors have addressed my concerns, and l  would like to retain my positive score.

**Claims And Evidence:**

The claims made in the submission are supported by clear and convincing evidence. The authors provided a solid theoretical foundation for their method, integrating Riemannian preconditioner and gate-value scaling to address key challenges.

**Essential References Not Discussed:**

The literature discussed by the authors is indeed comprehensive and closely related to the core topics addressed in the paper. They highlighted key advancements in LoRA, MoELoRA, and Riemannian preconditioning.

**Experimental Designs Or Analyses:**

I checked them. The experimental setup and analyses appear to be well-structured and appropriate for assessing the claims made, such as multiple benchmarks (Table 1/2/3), convergence analysis (Figure 2), and ablation study (Table 4.5).

**Methods And Evaluation Criteria:**

Methods and evaluation criteria are highly appropriate and well-suited for addressing the problem and application at hand, including benchmarks such as QA datasets, the GLUE benchmark, and VG/VMCBench.

**Other Comments Or Suggestions:**

While the theoretical contributions and experimental results are well-presented, it would be beneficial to include a more detailed discussion of the practical implications of this method.
And I highly recommend that the authors consider open-sourcing their implementation.

**Other Strengths And Weaknesses:**

Strengths:

* Nice presentation, Reasonable motivation, and Interesting theoretical contribution. The authors introduce a simple yet powerful idea inspired by mathematical principles.

* The authors propose a method based on gate scaling theory to enhance the performance of MoE-LoRA, which takes into account the influence of manifold curvature.

* The authors derive a rescaling method based on Riemannian preconditioning and provide a complete theoretical derivation process.

* This method effectively balances the gradient updates among experts, addressing challenges such as curvature distortion in the MoE.

* The engineering approximation seems to provide computational efficiency.

Weaknesses:

* The convergence is crucial for understanding this method. Regrettably, the authors did not carefully address this in Figure 2. A detailed explanation is necessary, such as the meaning of the dual-axis, and the significance of training loss and validation loss. Then, overlapping the axes in the middle of the subplots should be noted.

* Equation 13 appears highly valuable, yet its explanation could be more transparent. The authors should provide a clearer and more detailed interpretation of this equation.

* In Subsection 3.3, the authors propose a more flexible engineering approximation, which is an interesting contribution, this approach seems to achieve low computational overhead. However, more elaboration on the advantages of this approach would be helpful.

* A significant advantage of this method that enhances MoE-LoRA as a training strategy. However, the experiments comparing the proposed method with the MoE- LoRA baseline should be clearer. For example, MoLA-SGD (2, 4, 6, 8) should ideally be presented alongside MoLA in the lower part of the Table.

* The legend of Figure 3 should be streamlined for clarity.

**Questions For Authors:**

Please see Strengths And Weaknesses.

**Relation To Broader Scientific Literature:**

The paper is well-aligned with recent literature, like LoRA and LoRA Variants, MoELoRA, and Gradient Preconditioners. The authors discuss literature related to these concepts, including works such as MiLoRA, LoRA+, DoRA, MoLA, MoV, etc. And more, the authors have done a nice job of using theory to build connections between these concepts.

**Theoretical Claims:**

I reviewed the theoretical claims and their corresponding proofs in the submission. The authors provided detailed derivations and mathematical justifications for their methods, mainly focusing on the integration of Riemannian Preconditioner, such as Limitation 1/2 (Limited representation and Gradient Sub-optimality), Riemannian Preconditioner in LoRA Expert and Rescaling Preconditioners (Section 3.1/3.2).

---

> ### Author Rebuttal · Authors · 2025-04-01
>
> Thank you for providing **postive feedbacks on our presentations, derivations, and also experiments**. For the weaknesses and suggestions you pointed out, we highly value them and provide responses below:
>
> **Response to W1** about further explaning the convergence and fixing the issues in Figure 2
>
> We have added in our revision further explanations for the converging figures in Figure 2. For example, the x-axis represents training steps, the left y-axis in each figure represents the training or validation losses, while the right y-axis in each figure represents the accuracy metrics of test sets; Before implementing our gate-based rescaling method, the training and validation losses of RSGD optimizer across four tasks are significantly reduced around 100-200 steps, while after implementing our method, they are significantly reduced earlier around 0-100 steps; In addition to converging speed, we also notice an out-performance of our method in terms of converged loss and QA accuracy; Finally, to address the axis overlapping issue, we re-arranged each subplot in the figure to be less tight with each other and re-drew the Figure 2 in our revision.
>
> **Response to W2 and W3** about further elaborating Eq.13
>
> We further elaborate why we implement Eq.13 ($X=\hat{W}+\sum_{i=1}^{N_{Expert}}\hat{\sqrt{g_i}}B_iA_i+(g_i-\hat{\sqrt{g_i}})\hat{B_i}\hat{A_i}$), which is our engineering approximation for achieving Eq.11 and Eq.12. Since by forwarding as Eq.13, the gradient updating process of $X$ can be derived as the following (Similar as Eq.9 and Eq.10, we treat gate value $g_i$s as constants when focusing on the gradients of $A_i$s and $B_i$s):
> $$
> \begin{aligned}
> X_{new}=&\hat{W}+\sum_{i=1}^{N_{Expert}}[\hat{\sqrt{g_i}}(B_i-\eta\nabla_{B_i}\mathcal{L})(A_i-\eta\nabla_{A_i}\mathcal{L})+(g_i-\hat{\sqrt{g_i}})\hat{B_i}\hat{A_i}] \\\\
> =& (\hat{W}+\sum_{i=1}^{N_{Expert}}\hat{\sqrt{g_i}}B_iA_i+(g_i-\hat{\sqrt{g_i}})\hat{B_i}\hat{A_i})-\eta\sum_{i=1}^{N_{Expert}}\hat{\sqrt{g_i}}[B_i(\nabla_{A_i}\mathcal{L})+(\nabla_{B_i}\mathcal{L})A_i] \\\\
> =& X-\eta\sum_{i=1}^{N_{Expert}}\hat{\sqrt{g_i}}(B_i\nabla_{A_i}\mathcal{L}+\nabla_{B_i}\mathcal{L}A_i) \\\\
> =& X-\eta\sum_{i=1}^{N_{Expert}}(\hat{\sqrt{g_i}})^2Proj_{col(B_i)}(\nabla_{X}\mathcal{L})^T-\eta\sum_{i=1}^{N_{Expert}}(\hat{\sqrt{g_i}})^2Proj_{row(A_i)}(\nabla_{X}\mathcal{L}) \text{  (same as the derivation of Eq.10)} \\\\
> =& X-\eta\sum_{i=1}^{N_{Expert}}g_iProj_{col(B_i)}(\nabla_{X}\mathcal{L})^T-\eta\sum_{i=1}^{N_{Expert}}g_iProj_{row(A_i)}(\nabla_{X}\mathcal{L}),
> \end{aligned}
> $$
> so that Eq.12 can be achieved.
>
> The advantages of implementing Eq.13 can be elaborated from two aspects: Firstly, it achieves Eq.12 when doing gradient updating of $X$ while still keeps the original behavior of training gates, because it holds the same gate gradient, $\nabla_{g_i}X={A_i}^T{B_i}^T$, as normal forwarding; Secondly, it provides equivalent behavior and the same result of normal module forwarding $X=\hat{W}+\sum_{i=1}^{N_{Expert}}g_iB_iA_i$, and only requires a relatively low overhead.
>
> **Response to W4** about the presentation issue of MoLA experiments
>
> We found that the candidates order presented in our baseline comparison experiments (Table 4) may not appropriate. As you suggested, we have moved the $MoLA-SGD (2,4,6,8)$ candidate to the lower part of the table, alongside with other MoLA candidates.
>
> **Response to W5** about the legend of Figure 3
>
> As you mentioned, we notice that the legend of Figure 3 should be streamlined since it has a chance to cover part of the blue line (actually it does not cover, but that still leads to unclarity). As a result, we shortened the names of lines in the legend by deleting the "Loss" word and drew the figure again.
>
> **Response to your suggestion** of discussing practical implications
>
> Thank you for your suggestions. The practical implications of this work are mainly focused on boosting the training of MoE-LoRA, which may be applied to the fields like efficient and low-resource model training, continual or multi-task learning, stablized training and modular task adaptation, etc. For example, some current works use MoE-LoRA structure to distillate knowledge from a much larger dense model, such as Xu et al.[1] Our proposed method may enhance their distillation process. We have added these practical implications to our paper, as a new section before the conclusion section.
>
> **Response to your suggestion** about open-sourcing
>
> Yes, we will open-source our implementation on Github after this paper is published.
>
> [1] Xu, Haiyang, et al. "Sparse Mixture of Experts Language Models Excel in Knowledge Distillation." CCF International Conference on Natural Language Processing and Chinese Computing. Singapore: Springer Nature Singapore, 2024.

---

### Official Review · Reviewer_76YT · 2025-03-14

**Overall Recommendation:** 4

**Summary:**

This work proposes an improved training strategy for MoE-LoRA, aiming to address the limited representation and suboptimal gradient issues when fine-tuning foundation models with plain MoE-LoRA. They first analyze the limitations of LoRA, including the insufficient representation capacity of low-rank matrices and gradient optimization problems. To enhance the representation power of LoRA, they introduce the MoE framework and then incorporate Riemannian Preconditioners to optimize the gradient update process. Through theoretical analysis and experimental validation, they demonstrate the effectiveness of the improved method in various downstream tasks, including question answering, language understanding, and vision-language tasks.

**Claims And Evidence:**

Yes, the claims made in the submission are supported by clear and convincing evidence.

**Essential References Not Discussed:**

All key related works are discussed in the paper, to the best of my knowledge.

**Experimental Designs Or Analyses:**

Yes, I have reviewed the soundness and validity of the experimental designs and analyses.

**Methods And Evaluation Criteria:**

Yes, the proposed methods and evaluation criteria make sense for the problem or application at hand.

**Other Comments Or Suggestions:**

It is commendable that this work offers a nice theoretical depth. They provided rigorous derivations to demonstrate that this work can achieve full fine-tuning equivalency in mathematical way. Additionally, they provided robust engineering implementations and alternative approximations, making this method both practical and scalable. However, given that the manuscript involves a substantial number of formulas and derivations, it is strongly recommended that the authors carefully review each step of the derivations to ensure their rigor and accuracy.

**Other Strengths And Weaknesses:**

***Strengths,***
**Innovation and Theoretical Value**: The introduction of the Riemannian Preconditioner into the MoE-LoRA is an innovative approach. This work addressed the instability issues encountered during the training of plain MoE-LoRA. Moreover, the manuscript provides a detailed theoretical analysis of the gradient update process in MoE-LoRA, revealing the underlying problems. And they further propose a gradient rescaling method based on gating values, which offers a solid theoretical foundation for their method.
**Experiments and Performance**: The method has been extensively tested across a variety of downstream tasks, including question answering, GLUE benchmark tests, and vision-language tasks, using different base models such as Llama, GLM, and LLaVA. The results demonstrate the effectiveness and generalizability of the proposed approach. The improved MoE-LoRA achieves significant performance improvements when using base optimizers.
**Practicality**: The manuscript introduces an engineering approximation method, which decomposes the optimized and non-optimized parts in the forward propagation.  This approach effectively resolves the difficulties associated with directly implementing the theoretical method, making it practical. For me, I'm intrigued by this section.
**Flexibility**: This work can be seamlessly integrated into existing MoE-LoRA baselines, such as MoLA. It can serve as a theoretical complement to current MoE-LoRA training strategies.

***Weaknesses,***
**Notations issues**: Some necessary notations and operators should be declared before use, even if they are commonly used conventions. For instance, symbols like X and Proj. should be defined.
**Unclear Abbreviations**: Abbreviations should be fully explained, especially those that may not be universally understood. For instance, FFN should be clearly defined. Additionally, what is the meaning of FFT? Might this be a small typo, I guess?
**More explicit conclusion** (Equation 12): As far as I understand, Equation 12 appears to be the core conclusion of this work. Therefore, it is crucial to provide a clear and detailed explanation of how and why Equation 12 can achieve full fine-tuning equivalency.  This will help readers better understand the core contribution of this work.
**Analytical glitches**: n/k/r seem to be of significant importance. However, the analysis of these parameters appears to be insufficient. A more thorough investigation is needed, and the optimal candidates should be emphasized.

**Questions For Authors:**

After carefully reviewing the theoretical section, although I understand how the proposed training strategy achieves full fine-tuning equivalency, it would be highly beneficial for the manuscript if the authors could provide a clearer explanation of how and why Equation 12 enables full fine-tuning equivalency. This would greatly enhance the readability and comprehension of the core contribution.

**Relation To Broader Scientific Literature:**

The key contributions of the paper are well-grounded and significantly advance the broader scientific literature, particularly in LoRA, MoE, and optimization techniques for foundation models.

**Theoretical Claims:**

Yes, I carefully reviewed the theoretical proofs presented in the manuscript, particularly focusing on the core contributions related to the improved MoE-LoRA training strategy.

---

> ### Author Rebuttal · Authors · 2025-04-01
>
> Thank you for your **acknowledgment on our innovations and theoretical value**. We’ve checked our paper again carefully to address any issues you mentioned. Here are our responses to your valuable concerns:
>
> **Response to W1 and W2** about the notations and abbreviations issues
>
> We have carefully reviewed our paper again to address those undeclared or unclear notations and abbreviations. For example, as you mentioned, the symbol $X$ represents the overall weight matrix after integrating pretrained weights $W$ and LoRA modules $A$s and $B$s; $Proj_V(M)$ represents a projection function which projects a given matrix $M$ onto a sub space constructed by all vectors in set $V$. And when we treat all vectors in $V$ as the rows of another matrix, such as $P$, then the projecting action can be calculated as $Proj_{row(P)}(M) = MP^T(PP^T)^{-1}P$; FFN and FFT represent different concepts. FFN is the abbreviation for Feed-Forward Network; and FFT is the abbreviation for Fully Fine-Tuning. In our revision, we have addressed all the above notation issues, as well as some other unclear notation issues we found during our re-check.
>
> **Response to W3 and Q1** about more explanations of Eq.12 on its fully-finetuning equivalency
>
> Eq.12 is the refined Riemannian-preconditioned backpropagating equation in MoE cases, after applying our proposed gate-based rescaling method. It further approaches global fully finetuning because of two basic reasons: Firstly, it is derived by implementing Riemannian preconditioners to calibrate each LoRA expert's gradient (given by Eq.6), thus ensuring each LoRA expert can get close to their respective full-rank training behavior locally (i.e., per-expert fully-finetuning equivalency), according to Zhang et al.[1]; Secondly, we notice a further distortion of each expert space introduced by their respective gate values, leading to an inconsistency between per-expert local optimals and the global optimal. Therefore, we further introduce a respective gate value $g_i$ as a re-scaler to each expert's Riemannian preconditioner (given by Eq.11), to relieve the expert distortion resulting from the multiplication of gate value during forwarding. As a result, Eq.12 can further approach global fully-finetuning equivalency (e.g., larger gate values introduce less distortion. Thus, through Eq.12, experts with larger gate values are re-scaled less than those with smaller ones). We have already integrated above discussions about fully-finetuning equivalency in our revision.
>
> **Response to W4** about insufficient $n/k/r$ analysis
>
> We already presented our $n/k/r$ analysis in Table 5 in Section 4.6, which consists of seven different candidates tested under SGD and AdamW optimizers. Llama-3.2-3B serves as their foundation model. Table 5 already demonstrates our overall effectiveness across various $n/k/r$ configurations. To make the investigation further sufficient, we added two new experiments under LLaVA-v1.5-7B with two different $n/k/r$ configurations (16/8/4 and 10/5/4). Please refer to the results in our rebuttal to Reviewer 1wAK. An overall enhancement of our method can still be witnessed.
>
> We have already emphasized our preliminary conclusion in Section 4.6 that, the value of $k$ is more important to performance boosting of our method under SGD optimizers. Now we provide a further analysis about both our performance boosting and the final overall performance: Firstly, for performance boosting, we calculate its correlation with $n$, $k$ and $r$ from Table 5, which are $0.357$, $0.912$ and $0.093$ respectively under SGD optimizers, demonstrating our preliminary conclusion is correct. While under AdamW optimizers they are $0.001$, $0.197$ and $0.806$, indicating that $r$ might be more important for boosting AdamW; Secondly, for final performance, we claim that it results from both our boosting effectiveness and the MoE fundamental features (such as a too-large MoE structure may lead to underfitting, while a too-small one may lead to overfitting). After the mixed influence of both aspects, the optimal configuration for LLama-3.2-3B in our ScienceQA experiments is the $10/5/1$ with our re-scaled SGD optimization; while for LLaVA-v1.5-7B in VMCBench experiments, for example, it is $10/5/4$ with our re-scaled AdamW optimization.
>
> **Response to your suggestion** of reviewing each derivation step
>
> Thank you for your suggestion. We have reviewed all the derivation steps in our paper again to make sure each step is theoretically correct.
>
> [1] Zhang, Fangzhao, and Mert Pilanci. "Riemannian preconditioned lora for fine-tuning foundation models." arXiv preprint arXiv:2402.02347 (2024).

---

### Decision · Program_Chairs · 2025-05-01

**Decision:**

Accept (poster)

**Comment:**

This paper investigates the problem of fine-tuning foundation models using low-rank adapters (LoRAs). In particular, the authors identify that the mixture of LoRAs (MoE-LoRA) exhibits low robustness during both tuning and inference. To address this issue, they propose a new training strategy for MoE-LoRA that stabilizes and enhances its feature learning process through multi-space projections. Experiments on several datasets demonstrate the effectiveness of the proposed model.


Strengths:
1. The introduction of the Riemannian Preconditioner into MoE-LoRA is an innovative approach, and the authors further propose a gradient rescaling method based on gating values, which provides a solid theoretical foundation for their method.
2. The experiments are sufficient to demonstrate the effectiveness of the proposed model, with strong performance results.
3. The paper is well-written and easy to follow.


Weaknesses:
1. Some notations and terms are not clearly defined or consistently used, which may confuse readers.
2. Certain parts of the paper require further clarification, such as the convergence analysis in the figure and the usage of Equation 13.

Overall, this paper makes a valuable contribution to the field of training mixtures of LoRAs. I recommend the authors address the aforementioned issues as well as those pointed out by the reviewers, particularly in the camera-ready version, to improve clarity and ensure a more thorough explanation of key concepts.